# Multi-Epitope DC Vaccines with Melanoma Antigens for Immunotherapy of Melanoma

**DOI:** 10.3390/vaccines13040346

**Published:** 2025-03-25

**Authors:** Athanasios Seretis, Lukas Amon, Christoph H. Tripp, Giuseppe Cappellano, Florian Hornsteiner, Sophie Dieckmann, Janine Vierthaler, Daniela Ortner-Tobider, Markus Kanduth, Rita Steindl, Louis Boon, Joke M. M. den Haan, Christian H. K. Lehmann, Diana Dudziak, Patrizia Stoitzner

**Affiliations:** 1Department of Dermatology, Venereology and Allergology, Medical University of Innsbruck, 6020 Innsbruck, Austria; athanasios.seretis@uibk.ac.at (A.S.); christoph.tripp@i-med.ac.at (C.H.T.); florian.hornsteiner@i-med.ac.at (F.H.); sophie.dieckmann@i-med.ac.at (S.D.); daniela.ortner-tobider@i-med.ac.at (D.O.-T.);; 2Research Institute for Biomedical Aging Research, University of Innsbruck, 6020 Innsbruck, Austria; 3Laboratory of Dendritic Cell Biology, Department of Dermatology, University Hospital Erlangen, Friedrich-Alexander-University Erlangen-Nürnberg (FAU), 91052 Erlangen, Germany; lukas.amon@uk-erlangen.de (L.A.); christian.lehmann@uk-erlangen.de (C.H.K.L.); diana.dudziak@med.uni-jena.de (D.D.); 4Department of Health Sciences, Interdisciplinary Research Center of Autoimmune Diseases, Università del Piemonte Orientale, 28100 Novara, Italy; giuseppe.cappellano@med.uniupo.it; 5Center for Translational Research on Autoimmune and Allergic Disease-CAAD, Università del Piemonte Orientale, 28100 Novara, Italy; 6JJP Biologics, 00-728 Warsaw, Poland; louis.boon@jjp.bio; 7Department of Molecular Cell Biology and Immunology, Cancer Center Amsterdam, Amsterdam Institute for Immunology and Infectious Diseases, Amsterdam UMC Location Vrije Universiteit Amsterdam, 1081 HV Amsterdam, The Neatherlands; j.denhaan@amsterdamumc.nl; 8FAU I-MED, Friedrich-Alexander-University Erlangen-Nürnberg (FAU), 91052 Erlangen, Germany; 9Deutsches Zentrum Immuntherapie (DZI), 91054 Erlangen, Germany; 10Comprehensive Cancer Center Erlangen-European Metropolitan Area of Nuremberg (CCC ER-EMN), 91054 Erlangen, Germany; 11Department for Paediatrics and Adolescent Medicine, University Hospital Erlangen, 91054 Erlangen, Germany; 12Comprehensive Cancer Center Central Germany Jena/Leipzig (CCCG), 07743 Jena, Germany; 13Institute of Immunology, Jena University Hospital, 07743 Jena, Germany

**Keywords:** dendritic cells, C-type lectin receptors, immunotherapy, melanoma antigens, DC-targeted therapy

## Abstract

**Background/Objectives**: The revolution for the treatment of melanoma came with the approval of checkpoint blockade antibodies. However, a substantial proportion of patients show primary or secondary resistance to this type of immunotherapy, indicating the need for alternative therapeutic strategies. Dendritic cells (DCs) of the skin are prime targets for vaccination approaches due to their potential to prime naïve T cells and their accessibility. This study aimed to develop and evaluate novel vaccines targeting the C-type lectin receptor DEC-205 to deliver melanoma-associated antigenic peptides to skin DCs. **Methods**: We cloned MHC-I-restricted peptides from the glycoprotein (gp)100_25–33_ and Tyrosinase-related protein (trp)2_180–188_ into the DEC-205 antibody sequence with modified peptide cutting sites from the OVA_257–264_ SIINFEKL peptide. We tested their potential to induce CD8^+^ T cell responses in both in vitro and in vivo settings. Tumor growth inhibition was evaluated in the transplantable B16.OVA melanoma murine model using a multi-epitope DC-based vaccine combining both peptides. **Results**: The cross-presentation of both gp100 and trp2 peptides was confirmed in vivo when peptide sequences were flanked by the OVA_257–264_ peptide cutting sites. Moreover, the combination of both antigenic peptides into a multi-epitope DC vaccine was required to inhibit B16.OVA melanoma growth. **Conclusions**: Our findings suggest that a DC-targeted vaccination approach using multiple epitopes deriving from melanoma antigens could represent a promising strategy for melanoma therapy.

## 1. Introduction

Melanoma treatment has been revolutionized by immunotherapy with checkpoint inhibitors and tumor-targeted therapy with BRAF/MEK inhibitors. However, half of the patients still do not respond to therapy or develop primary or secondary resistance [1,2]. Even when both strategies are combined, response rates are not dramatically improved and severe side effects are observed in patients [3]. Therefore, there is still an urgent unmet clinical need to develop alternative approaches or novel combination therapies.

In contrast to checkpoint inhibition, clinical trials have demonstrated a high safety profile for dendritic cell (DC) vaccination in cancer immunotherapy [4,5,6,7]. The limited success of DC vaccination to date has been attributed to a variety of factors, including the laborious ex vivo generation of monocyte-derived DCs, the choice of tumor antigen, and the poor migration of DCs to lymph nodes draining the injection site [8,9,10]. For these reasons, the direct delivery of antigens to DCs by means of in vivo targeting is a more promising approach for DC vaccination. DC-targeting antibody–antigen fusion constructs have been first developed in Ralph Steinman’s lab in pre-clinical mouse models and have been proven to efficiently induce antigen-specific T cell responses [8]. The main receptors targeted are members of the endocytic C-type lectin receptor family [11,12,13,14]. One of the most used C-type lectin receptors for targeting is DEC-205, which guides antigens to endocytic vesicles, thereby promoting antigen presentation [12,15,16,17]. Antigen selection for delivery to DCs has focused mainly on model antigens, such as hen egg lysozyme, myelin-oligodendrocyte-glycoprotein, and ovalbumin (OVA) [16,18,19]. With regard to tumor immunotherapy, the targeted delivery of the OVA protein inhibited the growth of OVA-expressing melanoma [20,21]. However, endogenous tumor-associated antigens, such as gp100 or trp2, have been used in few studies, despite promising results in mouse tumor studies when targeting protein antigens with DEC-205 antibodies [22,23,24]. A recent clinical trial with DEC-205 conjugated to the tumor antigen NY-ESO1 reported the induction of T cell responses and in some cancer patients, stable disease [4]. In our study, we aimed at developing genetically engineered DEC-205 fusion proteins with peptide epitopes derived from melanoma-associated antigens.

Thus, we hypothesized that DEC-205-mediated DC vaccination using endogenous, melanoma-associated antigens could improve therapeutic outcomes in a pre-clinical melanoma mouse model. The use of whole proteins as antigens, their production, and their subsequent conjugation onto targeting antibodies presents several challenges, i.e., the large-scale production of the proteins themselves, the stoichiometry of conjugation onto the targeting antibody, and the potential allosteric inhibition of the antibody binding to its target. For this reason, we focused on short, MHC-I-restricted peptides derived from the melanoma-associated antigens gp100_25–33_ and trp2_180–188_. To enable presentation by DCs, we designed DEC-205 antibodies containing the relevant MHC-I peptides flanked by the five amino acid long sequences from the original protein or the five amino acid long flanking sequences of the OVA_257–264_ SIINFEKL peptide. Our results show the cross-presentation of gp100 and trp2 peptides in vivo and the induction of endogenous T cell responses against the targeted antigens. Finally, we demonstrated that combining both antigens into a multi-epitope DC vaccine, rather than single epitopes, slowed down tumor growth. Thus, vaccination with antibody-targeted multi-epitopes derived from commonly expressed melanoma antigens could be a promising approach as a future combination therapy for melanoma treatment.

## 2. Materials and Methods

### 2.1. Mice

Animal experiments were conducted in accordance with EU guidelines 86/609/EWG and national legal regulations (TVG 2012) and all efforts were made to minimize or avoid suffering. Experiments were approved by the Austrian Ministry of Science (66.011/0046-V/3b/2018, 2022-0.748.857). C57BL/6 breeders were purchased from Charles River Laboratories (Sulzfeld, Germany). Pmel-1 mice were kindly provided by Thomas Tüting (Otto-von-Guericke University, Magdeburg, Germany). Mice were housed and bred at the local animal facility of the Department of Dermatology, Venereology and Allergology of the Medical University of Innsbruck according to institutional guidelines.

### 2.2. DC-Vaccine Generation

For epitope selection, the MHC-I-restricted epitopes KVPRNQDWL and SVYDFFVWL of melanoma-associated peptides, gp100_25–33_ and trp2_180–188_, respectively, were previously described in [22,25,26]. To ensure efficient processing and presentation, we extended the 9 amino acid peptide sequence of both peptides at their N- and C-terminal ends. Two different variants of the 5 amino acid long flanking sequences were then cloned. Variant 1 contained the flanking 5 amino acids from the respective original protein (gp100_25–33_ or trp2_180–188_). Variant 2 contained the 5 amino acids flanking the OVA_257–264_ SIINFEKL peptide, as this peptide is well presented by DCs [27]. The respective antigenic sequences for both variants are shown in Figure 1A. The melanoma antigen peptide sequence for either gp100_25–33_ or trp2_180–188_ was genetically fused to the N-terminus of the heavy chain of the anti-DEC-205 antibody. As a negative control, the coding sequence of antigenic peptides were cloned into the isotype control (anti-TNP, clone: 7B4). To generate the combined multi-epitope DC vaccine, we cloned both MHC-I epitopes into the DEC-205 or isotype control antibody with the 5 amino acids flanking the OVA_257–264_ SIINFEKL peptide, as variant 2 proved to be more efficiently processed by DCs.

DC vaccines were produced as described earlier with some modifications [16,18]. Briefly, plasmids coding for the heavy and light chain of the DEC-205 were transiently transfected into HEK293T cells using calcium phosphate. Cell culture supernatants were harvested 7 days post-transfection and antibodies were purified by fast protein liquid chromatography (FPLC) using the NGC Discover 100 Pro Chromatography System (Bio-Rad, Hercules, CA, USA) and HiTrap Protein G HP columns (GE Healthcare, Chicago, IL, USA). LPS removal was performed by incubation with Triton X-114 (Sigma Aldrich, St. Louis, MO, USA). Endotoxin concentrations below 0.001 EU/μg were considered acceptable for the in vivo treatments, which was determined by the Toxin Sensor Chromogenic LAL assay kit (GenScript, Piscataway, NJ, USA). Protein concentration was estimated with the Pierce BCA Protein Assay kit (Thermo Scientific, Waltham, MA, USA). To verify vaccine production, 1 µg of antigen-antibody fusion products were blotted onto the nitrocellulose membrane (0.2 μm, Biorad Laboratories, Hercules, CA, USA) and incubated with goat anti-mouse IR dye 800CW (Li-Cor, Lincoln, NE, USA).

### 2.3. DEC-205 Binding Assays

Bone marrow-derived DCs (BMDCs) were generated as previously described [28]. Briefly, C57BL/6 bone marrow cells were harvested from femur and tibia bones and cultured for 6 days in RPMI 1640 (Lonza, Basel, Switzerland) supplemented with 10% fetal calf serum (FCS; PAN Biotech, Aidenbach, Germany), 2 mM L-glutamine (Lonza), 50 μM β-mercaptoethanol (Sigma), and 50 μg/mL Gentamycin (Gibco, Waltham, MA, USA), in the presence of GM-CSF 200 U/mL (Biozym, Hessisch Oldendorf, Germany). At day 6, BMDCs were incubated with 0.2 µg of DEC-gp100 and 0.2 µg of DEC-Trp2 antibodies for 15 min at 4 °C. A secondary rat anti-mouse antibody (clone RMG1-1, Biolegend, San Diego, CA, USA) was added and the PE signal intensity on BMDCs was analyzed by flow cytometry (Canto II, BD Biosciences, Franklin Lakes, NJ, USA). Residual binding sites were blocked using mouse γ-globulin (Jackson ImmunoResearch Laboratories, West Grove, PA, USA) for 15 min at 4 °C, before continuing with cell surface staining.

### 2.4. In Vivo Proliferation Assay

Spleens were harvested from gp100_25–33_-specific T cell receptor (TCR)-transgenic pmel-1 mice [26] and single cell suspensions were prepared. CD8^+^ T cells were enriched using the CD8a (clone Ly-2) T cell isolation kit (Miltenyi, North Rhine-Westphalia, Germany) according to the manufacturer’s instructions. For cell proliferation assays, purified CD8^+^ T cells were stained with 0.4 μM Carboxyfluorescein succinimidyl ester (CFSE, Invitrogen, Waltham, MA, USA) for 3 min at room temperature (RT). Then, 1 × 10^6^ CFSE-labelled CD8^+^ T cells were adoptively transferred intravenously (i.v.) into C57BL/6 mice. On the same day, mice were vaccinated as described below. Then, 3 days later, lymph nodes draining the vaccination site were collected and T cell proliferation was assessed by flow cytometry.

### 2.5. In Vivo DC-Based Vaccination

DC-targeting vaccines diluted in PBS at varying doses were subcutaneously (s.c.) injected into the right flank skin of the C57BL/6 mice. The Toll-like receptor (TLR)-ligand polyI:C (pI:C; 12.5 μg, Sigma Aldrich) and anti-mouse CD40 agonistic antibody (αCD40; 12.5 μg, clone: FGK45) were co-administered as an adjuvant. Vaccines plus adjuvants were injected weekly into the same flank skin site unless indicated otherwise.

### 2.6. Single Cell Suspension

Collected lymph nodes were mechanically disrupted and digested at 37 °C for 25 min using 300 μg/mL DNase I (Roche, Basel, Switzerland) and 250 μg/mL Collagenase D (Roche). After incubation, the reaction was stopped with 20 mM EDTA (Lonza) and the samples were pressed through a 40 μm cell strainer (Corning, Corning, NY, USA) to obtain single cell suspension.

### 2.7. Flow Cytometry

For dead cell exclusion, samples were stained with the fixable viability dye eFluor-780 (eBioscience, San Diego, CA, USA) at RT or 7AAD (BD Bioscience). Fc receptors on immune cells were blocked by anti-mouse CD16/CD32 antibody (clone 2.4G2, TONBO Biosciences, San Diego, CA, USA). Next, fluorescently labelled antibodies (Biolegend or BD Biosciences) were used: CD3 (clone 17A2), CD8 (clone 53–6.7), CD11c (clone HL3), CD11b (clone M1/70), CD19 (clone 1D3), NK1.1 (clone PK136), MHC-II (clone M5/114.15.2), CD103 (clone 2E7), CD45 (clone 30-F11), CD90.1 (clone OX-7), CD69 (clone H1.2F3), DEC-205 (clone NLDC-145)), CD62L (MEL-14), and CD44 (IM7). For staining, samples were incubated for 15 min at 4 °C.

Pentamer staining was performed on blood samples taken weekly on the day of vaccination using the gp100 (KVPRNQDWL, Proimmune, Oxford, UK) or trp2 pentamer (SVYDFFVWL, Proimmune) in PBS/0.5% BSA (Merck) for 20 min at RT. After incubation, cells were washed and stained for cell surface markers as described above. Samples were acquired on FACS Canto II (BD Biosciences) or Cytoflex S (Beckman Coulter, Brea, CA, USA) and analyzed using the Flowjo software (v10.10 BD Biosciences).

### 2.8. Transplantable B16.OVA Melanoma Model

B16.OVA tumor cells [29] were cultured in IMDM medium supplemented with 10% FCS (PAN Biotech), 50 μg/mL Gentamycin (Gibco), 0.05 mM β-mercaptoethanol (Sigma), and 1 mg/mL Paneticin G418 (PAN Biotech) to select for OVA-transfected tumor cells. Each mouse received 1.5 × 10^5^ B16.OVA cells in PBS s.c. into the left flank skin. The tumor size was measured using a digital caliper. The tumor area was calculated using the eclipse area calculation formula: π × (shortest diameter/2) × (longest diameter/2).

### 2.9. In Vitro Restimulation

Tumor-draining, inguinal lymph nodes were collected at day 23 after the first vaccination and single cell suspension was prepared by mechanical disruption. Lymph node cells were seeded into U-bottom, 96-well plates at 2 × 10^5^ cells/well in RPMI medium supplemented with 10% FCS, (PAN Biotech), 2 mM L-glutamine (Lonza), 50 μM β-mercaptoethanol (Sigma), and 50 μg/mL Gentamycin (Gibco). Cells were re-stimulated with 1 μg/mL of trp2 peptide (SVYDFFVWL, Think peptides) for 48 h. The culture supernatants were collected and secreted interferon (IFN) γ levels were detected by ELISA assay (Invitrogen).

### 2.10. Statistical Analysis

Flow cytometry data were analyzed with FlowJo software (BD Biosciences). All data are presented as median values ± standard deviation (SD). Statistical analyses were performed using GraphPad Prism software 10 (GraphPad Software, San Diego, CA, USA). Normality was assessed by the D’Agostino–Pearson omnibus normality test and Shapiro–Wilk normality test. Normally distributed data were analyzed using an unpaired t-test, one-way ordinary ANOVA, or two-way ANOVA as appropriate. Non-normally distributed data were analyzed using Mann–Whitney test or Kruskal–Wallis test, as appropriate. Only statistically significant comparisons are depicted in figures, unless otherwise stated. *p* < 0.05 was considered as statistically significant (*, *p* < 0.05; **, *p* < 0.01; ***, *p* < 0.001; ****, *p* < 0.0001).

## 3. Results

### 3.1. Cloning of gp100 and trp2 CD8^+^ T Cell Epitopes into DEC-205 Monoclonal Antibody

As previously described by us and others, DEC-205-antigen fusion constructs deliver antigens successfully to DC and induce antigen-specific T cell responses [18,30]. Hence, the melanoma antigen peptide sequence for either gp100_25–33_ or trp2_180–188_ was genetically fused to the N-terminus of the heavy chain of the anti-DEC-205 (clone NLDC-145) antibody, similarly to previously published work [16]. The human gp100 sequence (KLPRNQDWL) was selected due to its increased immunogenicity compared to the mouse gp100 sequence [31]. The five amino acid long flanking sequences derived from the respective melanoma antigens themselves (var1), or the flanking sequences of the OVA_257–264_ SIINFEKL peptide (var2), were used to ensure antigen processing (Figure 1A). As a negative control, the coding sequence of antigenic peptides were cloned into an anti-TNP isotype control antibody (clone: 7B4). The Western blot analysis of FPLC-purified antibody–antigen constructs confirmed the production of both heavy and light antibody chains, with a size of 50 kDa and 25 kDa, respectively (Figure 1B, Appendix A).

To assess the binding specificity of our antibody–peptide constructs to their designated target DEC-205 on DC, we generated BMDCs from C57BL/6 mice and on day 6 incubated the immature BMDCs with the DEC-gp100 or the DEC-Trp2 vaccines. A secondary PE-labelled rat anti-mouse IgG was used to detect the binding of the antibody–peptide constructs to BMDCs as analyzed by flow cytometry. Isotype constructs were used as negative controls to detect unspecific binding. The binding capacity of all DEC-gp100 and DEC-Trp2 variants was similar compared to a commercially available DEC-205 antibody (Figure 1C, Appendix A).

Thus, DEC-gp100 and DEC-Trp2 fusion proteins were successfully produced and assembled to DC-targeting antibody–antigen constructs.

**Figure 1 vaccines-13-00346-f001:**
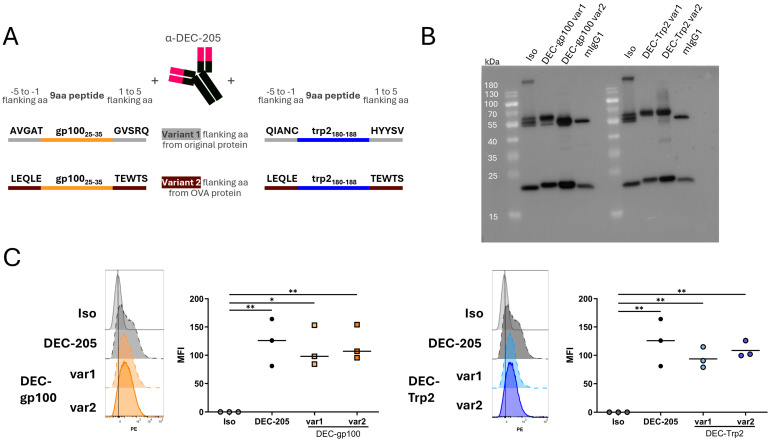
Cloning strategy and verification for antibody–antigen fusion proteins. (**A**) Two variants (var) were designed for each vaccine. Var1 contained the 5 amino acids (aa) flanking both sides of the respective peptides in the melanoma antigen protein gp100_25–35_ or trp2_180–188_, whereas var2 contained the 5 aa flanking both sides of the OVA_257–264_ SIINFEKL peptide. (**B**) Western blot of the DEC-gp100 and DEC-Trp2 productions. (**C**) Day 6 immature BMDCs were incubated with 0.2 µg of in-house-produced isotype control (Iso), commercially available DEC-205-PE (DEC-205), or each of the in-house-produced DC vaccines, DEC-gp100 (left side) or DEC-Trp2 (right side), before visualization by fluorescently labelled rat anti-mouse antibody, one-way ANOVA with Tukey’s multiple comparisons test. Representative histograms and results for 3 independent experiments are shown (*, *p* < 0.05; **, *p* < 0.01).

### 3.2. Vaccine Variants with OVA_257–264_ SIINFEKL Peptide Flanking Sites Are Cross-Presented Better to CD8^+^ T Cells

To evaluate the cross-presentation of DC vaccines in vivo, we adoptively transferred 1 × 10^6^ CFSE-labelled gp100-specific CD8^+^ T cells into C57BL/6 mice and subsequently vaccinated the mice subcutaneously with the DEC-gp100 variants into the flank skin. The isotype control (Iso-gp100) or unconjugated gp100_25–33_ peptide were used as negative and positive controls, respectively. An adjuvant mix of polyI:C together with the agonistic anti-CD40 antibody (pI:C/αCD40) was co-administered together with the DC vaccines or the controls, as described earlier. On day 3 after vaccination, inguinal lymph nodes were collected, and single cell suspensions were analyzed by flow cytometry (experimental design, Figure 2A). Transferred cells were identified as CD3^+^CD8^+^CD90.1^+^ T cells and CFSE dilution, indicative of T cell proliferation, was detected by flow cytometry (gating strategy shown in Appendix A). We observed a higher T cell stimulation with variant 2 containing the OVA_257–264_ SIINFEKL peptide flanking regions (Figure 2B). In detail, DEC-gp100 var2 induced superior T cell proliferation at a concentration of 0.5 µg, indicative of improved antigen processing and presentation by DCs when compared to DEC-gp100 var1 (Figure 2B). We concluded that in this setting, the efficacy of the DEC-gp100 var2 vaccine was similar in comparison with the free gp100 peptide, which was used as the positive control; therefore, it was equally as effective as antibody–antigen constructs from other publications [17].

The cross-presentation of the DEC-Trp2 vaccine had to be tested by an in vivo vaccination approach because of the lack of a TCR transgenic mouse model. Therefore, we immunized C57BL/6 mice with the DEC-Trp2-variant 1 or -variant 2 plus pI:C/αCD40 for seven weeks in a weekly interval. Afterwards, draining lymph node cells were restimulated with the trp2_180–188_ peptide and IFN-γ levels in the supernatants were measured by ELISA 48 h later (experimental design, Figure 2C). Only lymph nodes from mice immunized with variant 2 had detectable IFN-γ levels; therefore, variant 2 was used for further experiments.

All together, these data demonstrate that the variant 2 containing the flanking five amino acid sequences of the OVA_257–264_ SIINFEKL peptide was processed in a superior way than then the original sequences in the gp100 and trp2 proteins, thus resulting in proper peptide antigen presentation to CD8^+^ T cells in vivo.

### 3.3. Cloning of DEC-205 Multi-Epitope Vaccine with gp100 and trp2 Peptides

After verifying that variant 2 of both single epitope DC vaccines was able to induce the cross-presentation of either the gp100 or the trp2 peptides in vivo in a superior way than variant 1, we proceeded to cloning both epitopes in a multi-epitope DEC-205 antibody. Therefore, the variant 2 sequences of both peptides, gp100_25–33_ and trp2_180–188_, were cloned into the heavy chain of the DEC-205 antibody (Figure 3A). After the production and purification of the multi-epitope vaccine, we confirmed the presence of both heavy and light chains by the Western blot analysis. The FPLC-purified protein was identified as a mouse antibody with the expected molecular weights of heavy and light chains (Figure 3B, Appendix A). The binding of DEC-gp100-Trp2 to immature day 6 BMDCs confirmed that the antibody retained an equal binding efficiency to the DEC-gp100 and commercial DEC-205 antibodies (Figure 3C). As DEC-gp100 demonstrated similar binding efficiency to DEC-Trp2 (Figure 1C), DEC-gp100 was used to validate the efficacy of the multi-epitope vaccine.

To verify that both antigens were able to be cross-presented in vivo, we vaccinated C57BL/6 mice following a scheme of weekly vaccinations using the DEC-gp100-Trp2 DC vaccine or a multi-epitope isotype control (Iso-gp100-Trp2) in the presence of pI:C/αCD40 (experimental design, Figure 4A). Parallel blood samples were collected on the same day and antigen-specific T cells were detected using pentamer staining. As shown in Figure 4B, the multi-epitope DC vaccine was capable of promoting the expansion of both gp100- and trp2-specific CD8^+^ T cell populations over a 5 week-period (Figure 4B, Appendix A). Furthermore, we asked if the efficacy of the cross-presentation was affected by the presence of two peptides in close proximity. To this end, we adoptively transferred CFSE-labelled, gp100-specific CD8^+^ T cells into C57BL/6 mice and subsequently vaccinated the mice using the single epitope DEC-gp100, the multi-epitope DEC-gp100-Trp2, or the multi-epitope isotype control in the presence of pI:C/αCD40. On day 3 post-vaccination, vaccination-site-draining lymph nodes were collected and analyzed by flow cytometry (experimental design, Figure 4C). We observed that gp100-specific CD8^+^ T cells proliferated equally well in both DEC-gp100 and DEC-gp100-Trp2 immunized mice (Figure 4D).

Taken together, these data confirm that the multi-epitope DC vaccine DEC-gp100-Trp2 could be successfully produced and was able to induce the cross-presentation of both antigenic epitopes, promoting the expansion of distinct, antigen-specific CD8^+^ T cell populations. Moreover, the cross-presentation of the peptides is not hindered by their proximity but is similarly efficient.

### 3.4. DEC-gp100-Trp2 Slows Down Against B16-OVA Melanoma Growth by Inducing Memory T Cells

After we confirmed the induction of CD8^+^ T cell responses after vaccination with the multi-epitope DEC-205 antibody, we tested this vaccine in a prophylactic melanoma therapy approach. C57BL/6 mice were immunized either with the DEC-gp100-Trp2 or with one of the single-epitope DC vaccines, DEC-gp100 or DEC-Trp2. The DC vaccines including the isotype control containing both epitopes (Iso-gp100-Trp2) were injected in the presence of pI:C/αCD40. At the same time, unvaccinated control mice were injected with saline (PBS). Two days after the second immunization, we transplanted B16.OVA cells into the opposite flank from the vaccination site. Vaccinations were continued weekly until the experimental endpoint (experimental design, Figure 5A). Tumor growth measurements showed that the DEC-gp100-Trp2 vaccine could delay tumor outgrowth compared to multi-epitope isotype- or single-epitope-vaccine-immunized mice (Figure 5B). The survival of vaccinated mice was significantly improved by DEC-gp100-Trp2 immunization compared to the other treatment groups or the control group (Figure 5C) [32]. To confirm that the inhibition of tumor growth was due to the expansion of melanoma antigen-specific CD8^+^T cells after vaccination with the multi-epitope DC vaccines, C57BL/6 mice were immunized with the multi-epitope DC vaccine, DEC-gp100-Trp2, or with one of the single-epitope DC vaccines, DEC-gp100 or DEC-Trp2. Control treatment groups received Iso-gp100-Trp2, DEC-OVA, or were left untreated. All treatment groups received pI:C/αCD40, except for the untreated, which received PBS. On day 9 post-immunization, the mice were challenged with B16.OVA cells (experimental design, Figure 5A). Immunizations were repeated weekly, blood samples were collected on the same day, and antigen-specific T cells were detected using gp100_25–33_ and trp_180–188_ specific pentamer staining. As shown in Appendix A, immunization using single-epitope DC vaccines induced the expansion of CD8^+^ T cells specific for the corresponding epitope. The expansion of both gp100- and trp2-specific CD8^+^ T cells was induced after immunization with the multi-epitope DC vaccine DEC-gp100-Trp2. Moreover, immunization with the DEC-OVA DC vaccine was unable to induce gp100- or trp2-specific CD8^+^ T cells (Appendix A). These data indicate that the expansion of antigen-specific CD8^+^ T cells in the periphery is restricted to the epitope contained in the DC vaccine.

Finally, we investigated the activation pattern of endogenous CD4^+^ and CD8^+^ T cells in the tumor-draining lymph nodes at week 4 (experimental design, Figure 5D). The week 4 time point was specifically chosen to ensure that all animals remained alive, regardless of the treatment received. Tumor growth was monitored and the inhibition of tumor growth by the multi-epitope DC vaccine was confirmed (Appendix A). Tumor-draining lymph node cells were analyzed for viable CD45^+^CD3^+^CD8^+^ T cells (gating strategy shown in Appendix A). CD8^+^ T cells were identified as either CD44^+^ single positive effector memory T cells (EM) or CD62L^+^CD44^+^-double positive central memory (CM) T cells. Central and effector memory CD8^+^ T cells were significantly increased in mice vaccinated with DEC-gp100-Trp2 compared to multi-epitope isotype-control mice (Figure 5E). A similar induction of CM and EM T cells was also evident in the CD4^+^ T cell compartment in DEC-gp100-Trp2-treated mice (Figure 5E).

Taken together, these results demonstrate that the multi-epitope DEC-205 DC vaccine is capable of inducing an immune response strong enough to delay tumor growth in the B16.OVA model. Furthermore, we have evidence that the T cell response is not restricted to the antigens used but rather spreads to include a CD4^+^ T cell response as well.

## 4. Discussion

In this study, we assessed the potential of DC-targeting treatments as a therapeutic approach for melanoma treatment [33]. For this purpose, we chose to deliver antigens to DCs in vivo using a DEC-205 targeting antibody, as previously described [15,16,17,34,35]. Most of these studies have focused on model or viral antigens, such as OVA [12,15,16,17,19,21,32,34,36] or long protein sequences of tumor-associated antigens [22,23,37,38]. As long protein sequences could potentially interfere with the structure of the antibody or binding to the target receptor [39], we chose to use short MHC-I-restricted peptide sequences. In our study, we show the potential of designing multi-epitope peptide DC vaccines for the treatment of melanoma. Our results highlight that the cloning strategy is essential to ensure proper peptide processing and antigen presentation on MHC-I as the use of OVA_257-264_ SIINFEKL peptide cutting sites proved superior over those from the original protein. Moreover, the design of multi-epitope vaccines demonstrated more potent effects on tumor growth in a transplanted B16.OVA melanoma mouse model. Overall, DEC-205-targeted multi-peptide vaccines derived from commonly expressed melanoma antigens could be a promising approach for a future combination therapy for melanoma.

DCs are an attractive choice for cancer immunotherapy. The possible application strategies of DC immunotherapy are numerous and clinical trials have demonstrated their safety and tolerability in the past [33]. The selection of appropriate antigens for DC immunotherapy is of outmost importance. Neoantigens are an alluring choice due to their high specificity for tumor cells and increased immunogenicity. However, neoantigen identification can be time consuming and is limited to each patient. Alternatively, tumor-associated antigens could provide an off-the-shelve vaccine choice, e.g., peptides selected for their expression and immunogenicity would allow for immediate use on patients. Our findings are in line with previous studies showing that cross-presentation in contrast to direct antigen presentation requires flanking sites with specific qualities [40]. Interestingly, in the same study, longer flanking sites (35 aa N-terminal sites, 23 aa C-terminal site) were reported to be necessary for optimal cross-presentation (39). However, in our vaccine constructs, peptide flanking sites of just five amino acids were sufficient for successful cross-presentation when the OVA_257–264_ SIINFEKL peptide cutting sites were used. This raises the question of whether the cross-presentation efficiency of our vaccine constructs could be improved by the elongation of the flanking sites of the peptide sequences. Nonetheless, extending the flanking regions could result in a longer fusion construct, which could hinder the folding and binding capacity of the DEC-205 antibody. On the other hand, as the antigenic peptides in our DC vaccine are routed into DCs through DEC-205-mediated uptake, this might already facilitate the cross-presentation of the fused peptide construct [15,17,18]. Given that DEC-205 acts as a scavenger receptor for apoptotic material [41], it is very likely that this mechanism contributes to the routing of the peptides to MHC-I molecules [15,17,18].

We show in our setting that the DC-targeting treatment delays tumor growth in a prophylactic setting. It remains to be investigated if our DC-targeting vaccines would also work in a therapeutic setting. It is striking that although DC-targeting vaccines have been studied for many years, their clinical success is very limited [14]. However, recent studies have demonstrated that DCs are pivotal for the success of cancer immunotherapy, including checkpoint blockade antibody therapy [42,43]. An explanation for the lack of clinical success of DC vaccines is the strong immunosuppressive tumor microenvironment with the infiltration of regulatory T cells, suppressive myeloid cells, high expression of inhibitory molecules, and down-regulation of tumor antigens [44,45]. These mechanisms allow tumors to escape immune surveillance. It is therefore tempting to suggest that DC-targeting vaccines could be used in a combination with other immunotherapies or tumor-targeted therapies to boost tumor immunity [46,47].

Moreover, our results show that the combination of short peptide sequences in a DC-targeting vaccine can induce the expansion of antigen-specific CD8^+^ T cells in the periphery. Interestingly, peripheral CD8^+^ T cells were increased, depending on the epitope contained in the DC vaccine. However, this expansion seemed to be broader in the tumor microenvironment and was not limited to CD8^+^ T cells only but included CD4^+^ T cells, possibly through antigen spreading, despite the lack of an MHC-II-restricted antigen in our vaccine design. Thus, DC vaccination could act at multiple levels, by increasing the pool of antigen-specific T cells but also by broadening the antigenic repertoire of tumor-infiltrating T cells. This multi-level effect of DC vaccination could be beneficial in combination with other immunotherapy approaches, such as boosting the T cell response in conjunction with checkpoint blockade antibodies in the case of primary or secondary resistance to immunotherapy. Translation into the patient situation is another challenge for the future due the highly polymorphic state of the human HLA locus; however, there are predominant alleles that cover the majority of the population, which could be used for peptide selection [48].

## 5. Conclusions

In conclusion, our findings show for the first time that robust anti-tumor responses can be achieved by combining short-length, tumor-associated antigens in a DC-targeting vaccine. Further investigation is needed to expand on our multi-epitope DC-vaccine system, including additional MHC-I- and MHC-II-restricted peptides to further boost anti-tumor immune responses.

## Figures and Tables

**Figure 2 vaccines-13-00346-f002:**
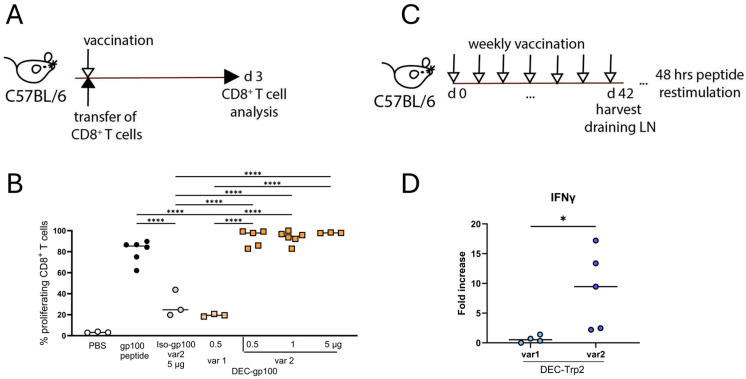
Comparison of DEC-205 DC vaccine variants by in vivo cross-presentation assays. (**A**,**B**) C57BL/6 mice received 1 × 10^6^ purified, CFSE-labelled gp100-specific CD8^+^ T cells intravenously prior to subcutaneous vaccination with 0.5 µg DEC-gp100-var 1, 0.5, 1, or 5 μg of DEC-gp100-var 2, 5 µg Iso-gp100, 5 µg gp100 peptide, or vehicle (PBS) control. All treatment groups except vehicle (PBS) control received the adjuvant mix pI:C/aCD40 (12.5 μg each). T cell proliferation was determined by flow cytometry analysis of the vaccination-site-draining lymph node on day 3 post-vaccination. (**A**) Experimental design and (**B**) results for 3 independent experiments are shown, one-way ANOVA with Tukey’s multiple comparisons test, n = 3–6 mice/group. (**C**,**D**) C57BL/6 mice were vaccinated weekly by subcutaneous injections of 5 μg of DEC-Trp2-var1 or -var2 in the presence of pI:C/aCD40 (12.5 μg each) for 7 weeks. At endpoint, vaccination-site-draining lymph nodes were collected, and cell suspensions were restimulated ex vivo with 1 μg/mL trp2 peptide for 48 h. (**C**) Experimental design. (**D**) ELISA results showing fold change of IFN-γ release trp2- over no peptide-restimulated cells. Unpaired *t* test, n = 4–5 mice (*, *p* < 0.05; ****, *p* < 0.0001).

**Figure 3 vaccines-13-00346-f003:**
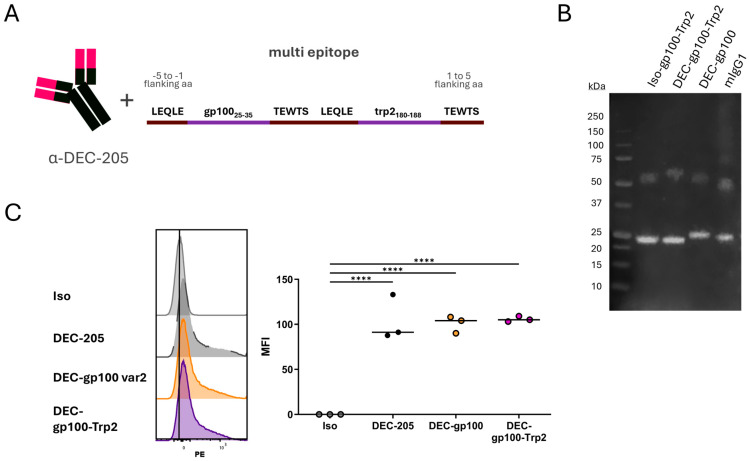
Cloning strategy and verification of the DEC-gp100-Trp2 multi-epitope DC vaccine. (**A**) Both peptides for gp100_25–33_ and trp2_180–188_ were cloned into the heavy chain of the DEC-205 antibody. Each peptide was flanked by the 5 amino-acid (aa) flanking region of the OVA_257–264_ SIINFEKL peptide (var2). (**B**) Western blot verification of the purification of both heavy and light chains of the DEC-gp100-Trp2 antibody and its isotype control (Iso-gp100-Trp2). (**C**) 6-day immature BMDCs were incubated with 0.2 μg in-house-produced multi-epitope isotype control (Iso), commercially available DEC-205-PE (DEC-205), DEC-gp100, or DEC-gp100-Trp2. Antibody binding was visualized by fluorescently labelled rat anti-mouse antibody, one-way ANOVA with Tukey’s multiple comparisons test. Representative histograms and results of 3 independent experiments are shown (****, *p* < 0.0001).

**Figure 4 vaccines-13-00346-f004:**
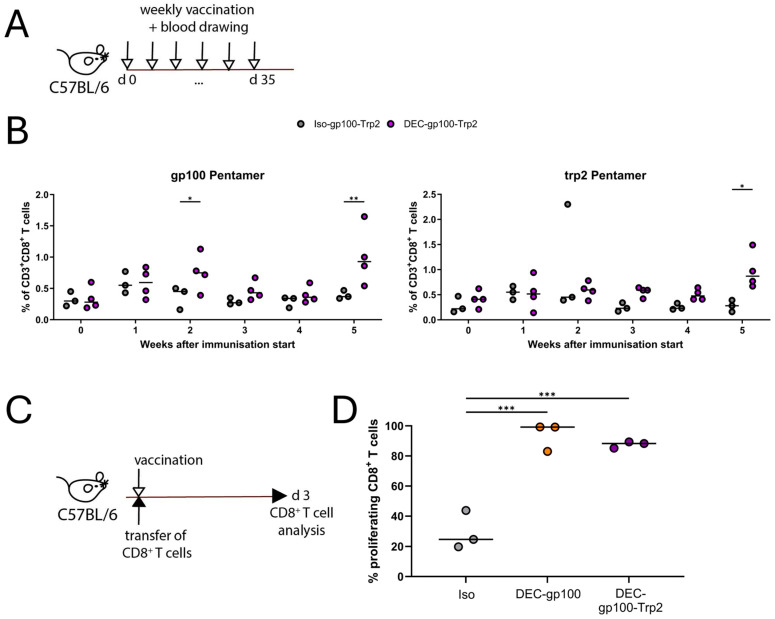
In vivo testing of the multi-epitope DEC-gp100-Trp2 DC vaccine. (**A**,**B**) C57BL/6 mice received subcutaneous vaccinations into the flank skin with 5 μg of the DEC-gp100-Trp2 multi-epitope DC vaccine or multi-epitope isotype control in the presence of pI:C/aCD40 (12.5 μg each) adjuvant mix. Blood samples from the mice were collected weekly on the same day as vaccination and blood samples were analyzed via flow cytometry for the presence of antigen-specific T cells. (**A**) Experimental design. (**B**) Results showing the expansion of endogenous antigen-specific CD8^+^ T cells for each of the pentamers tested. One experiment is shown, two-way ANOVA with Tukey’s multiple comparisons test. n = 3–4 mice. (**C**,**D**) C57BL/6 mice received 1 × 10^6^ CFSE-labelled, purified gp100-specific CD8^+^ T cells intravenously prior to subcutaneous vaccination with 0.5 µg DEC-gp100, 0.5 μg DEC-gp100-Trp2, or 0.5 µg Iso-gp100. All treatment groups received the adjuvant mix of pI:C/aCD40 (12.5 μg each). T cell proliferation was determined by flow cytometry analysis of the vaccination-site-draining lymph node on day 3 post-vaccination. (**C**) Experimental design. (**D**) Results showing gp100-specific T cell proliferation, one-way ANOVA with Tukey’s multiple comparisons test. n = 3 (*, *p* < 0.05; **, *p* < 0.01; ***, *p* < 0.001).

**Figure 5 vaccines-13-00346-f005:**
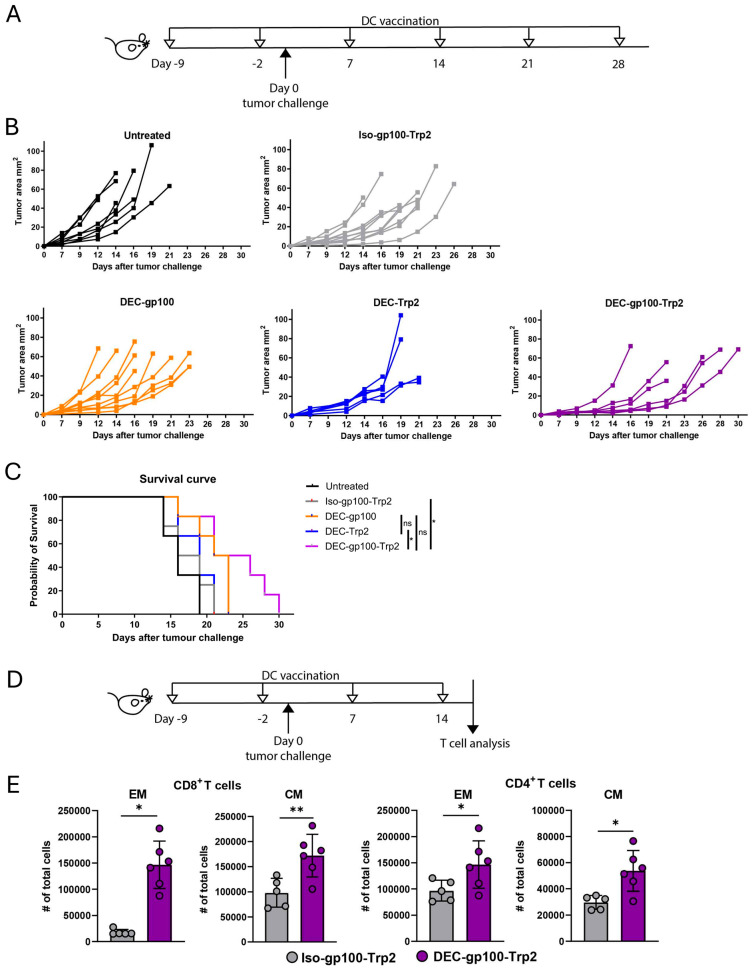
Multi-epitope DC vaccine delays significantly the growth of B16.OVA melanoma. C57BL/6 mice were vaccinated weekly with subcutaneous injections into the right flank skin of 5 μg of Iso-gp100-Trp2, DEC-gp100, DEC-Trp2, or DEC-gp100-Trp2 DC vaccine in the presence of pI:C/aCD40 (12.5 μg each), or by subcutaneous injections of PBS control (untreated). On day 9, mice were transplanted with 1.5 × 10^5^ B16.OVA cells by subcutaneous injection into the opposite (left) flank skin (**A**) Scheme of experimental design. (**B**) Tumor growth curves for every single mouse and (**C**) Kaplan–Meyer survival curves are shown curve comparison performed via Mantel–Cox test. Each line represents 1 mouse, and 2 experiments are shown. n = 4–9 mice. (**D**) Scheme of experimental design. (**E**) C57BL/6 mice were vaccinated with DEC-gp100-Trp2 DC-based vaccine and isotype control Iso-gp100-Trp2 and challenged with B16.OVA cells on day 9 post-immunization. On day 23 of the experiment, tumor-draining lymph nodes were analyzed for endogenous CD4^+^ and CD8^+^ T cell activation as assessed by flow cytometry. Unpaired t test, n = 5–6 mice (*, *p* < 0.05; **, *p* < 0.01).

## Data Availability

The data supporting the findings of this study are available from the corresponding author, [P.S.], upon reasonable request.

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
