# Peer review of "Multi-Epitope DC Vaccines with Melanoma Antigens for Immunotherapy of Melanoma"

_vaccines, 2025, doi:10.3390/vaccines13040346_

Round 1
Reviewer 1 Report
Comments and Suggestions for Authors
This is an interesting manuscript that describes the development of the DEC-205 antibody fused to MHC-I-restricted peptides from the glyco-34 protein (gp)100 and tyrosinase-related protein (trp)2 flanked by peptides from OVA protein. Authors, also find that the combination of these peptides in DC-based vaccination is more efficient in inhibiting melanoma tumor growth in vaccinated mice transplanted with B16.OVA cells.
The paper is properly introduced and the methods clearly describe the construct generation and all in vitro and in vivo experiments done.
However, this reviewer has some observations that could be useful to improve the work.
Major comments
Comment 1: the authors should add the statistics analysis and significance in the presentation of all results and the legends for *, **, *** symbols. This is particularly important for the Figure 4B, where it is very hard to establish the significant difference between isotype and DEC-205 constructs at the different timepoints. Also in the Figure 5C, it would be important to specify the statistical difference between the vaccination with single and multiple peptides.
Comment 2: Can authors explain why they decided to transplant B16.OVA melanoma cells and not B16 melanoma cells? The use of OVA peptides at flanking sites of gp100 and Trp2 petides can obviously work to potentiate the immune response against specific melanoma peptides, but the use of B16.OVA cells can create a bias: how the authors can be sure that the tumor growth inhibition was due to the in vitro observed immune response specific for melanoma peptides and not against OVA peptides? Experiments described in Figure 2 demonstrate the induction of melanoma-specific responses with the var2 of the construct, but how the authors can be sure that it is sufficient to kill transplanted melanoma cells and it is not overcome by the response against OVA peptide?
In this regards, can the authors add an experiment with mice transplanted with B16 melanoma cells or better analyze the specificity of CD4 and CD8 T cells from tumor-draining lymphonodes from mice translpanted with B16.OVA cells (for example with pentamers)? Alternatively, It could be useful to have the DEC-205 antibody fused only with OVA pentides as further control.
Minor comments
Comment 1: Authors need to correct the legend in figure 4B from “Iso-gp100” to “Iso-gp100-Trp2”.
Comment 2: The Figure 5 legend lacks of the point E description, or, better, the point D seems the description of the 5E figure, and not of 5D.
Author Response
Major Comments:
- The authors should add the statistics analysis and significance in the presentation of all results and the legends for *, **, *** symbols. This is particularly important for the Figure 4B, where it is very hard to establish the significant difference between isotype and DEC-205 constructs at the different timepoints.
Response #1: We apologize for the missing information. The statistical analysis section in methods has been altered to include the legends for *, **, *** symbols (lines 203-208). Additional information has been added in the same section regarding the statistical tests used. Figure legends have also been adapted to include statistical tests performed on the data sets and the appropriate symbols were added to the figures to indicate significance where applicable.
- Can authors explain why they decided to transplant B16.OVA melanoma cells and not B16 melanoma cells? The use of OVA peptides at flanking sites of gp100 and Trp2 peptides can obviously work to potentiate the immune response against specific melanoma peptides, but the use of B16.OVA cells can create a bias: how the authors can be sure that the tumor growth inhibition was due to the in vitro observed immune response specific for melanoma peptides and not against OVA peptides? Experiments described in Figure 2 demonstrate the induction of melanoma-specific responses with the var2 of the construct, but how the authors can be sure that it is sufficient to kill transplanted melanoma cells and it is not overcome by the response against OVA peptide?
Response #2: We thank the reviewer for raising this question as this makes us aware that the vaccine design was not properly explained in the manuscript. We actually used the 5 amino acids flanking both sides of the OVA257-264 SIINFEKL peptide, not the OVA257-264 SIINFEKL peptide itself. Thus, this peptide is never produced by the transiently transfected HEK293T cells. We added some information to the manuscript in the introduction (lines 90-91), figure legend 1 (lines 238) and results part (lines 257).
- In this regards, can the authors add an experiment with mice transplanted with B16 melanoma cells or better analyse the specificity of CD4 and CD8 T cells from tumor-draining lymph nodes form mice transplanted with B16.OVA cells (for example with pentamers)? Alternatively, it could be useful to have the DEC-205 antibody fused only with OVA peptides as further control.
Response #3: In response to a similar question raised reviewer 2 on the melanoma-antigen specificity of the induced T cell response, we included data in the revised manuscript on the expansion of melanoma antigen-specific T cells in the periphery of immunized tumor-bearing mice (see new Supplementary Figure S5). The gp100- or trp2-specific CD8+T cells were detected in the blood of mice that received the DC-based vaccines containing one or both melanoma epitopes. In addition, DEC-OVA-immunized mice showed no significant increase of gp100- or Trp2-specific CD8+ T cells, indicating that the expansion of T cells in the periphery was dependent upon the DC-based melanoma vaccines. The description of the new data can be found in the results part (lines 364-380) and the discussion (lines 461-464, 467-469) of the revised manuscript.
Although these data suggest that the DC-based vaccines expand antigen-specific T cells in the periphery of the mice, initiating an anti-tumor immune response, we cannot exclude the presence of other antigen-specific T cells in the tumor microenvironment that could respond to inflammatory stimuli and contribute to the anti-tumor immune responses observed, potentially by epitope spreading. Our analysis of effector and central memory CD4+ T cells in the tumor microenvironment seems to support this idea.
Minor comments
- Authors need to correct the legend in figure 4B from “Iso-gp100” to “Iso-gp100-Trp2”.
Response #4: We apologize for the wrong information in the figure legend which has been corrected accordingly and thank the reviewer for bringing this to our attention.
- The Figure 5 legend lacks of the point E description, or, better, the point D seems the description of the 5E figure, and not of 5D.
Response #5: We apologize for the incomplete figure legend and thank the reviewer from bringing this to our attention. The description of Figure 5D and 5E has been corrected accordingly (lines 386-391).
Reviewer 2 Report
Comments and Suggestions for Authors
The work of Seretis et al. is an interesting PoC study demonstrating the possibility of targeting dendritic cells through the C-type lectin receptor DEC-205 in order to improve immunogenicity of vaccines.
There is only one major issue that I’d like to raise. The description of results of the crucial in vivo experiment should be expanded with data on the presence of T cells specific to epitopes included into the vaccine. Otherwise, it is hard to justify that the tumor growth inhibition is caused by the antigen-specific immune response and not the non-specific activation of dendritic cells. Alternatively, the DEC-205 mAb should be used as a control.
The potential of therapeutic application of this approach can be limited due to the challenges of manufacturing mAb-like molecules fused with epitope-containing peptide chains. Therefore, data on expression level and protein purity would be useful.
The use of such vaccines will be limited by patients with particular MHC allele. It’d be interesting to know authors’ ideas on how to deal with this limitation.
Author Response
Comments and suggestions for Authors
The work of Seretis et al. is an interesting PoC study demonstrating the possibility of targeting dendritic cells though the C-type lectin receptor DEC-205 in order to improve immunogenicity of vaccines.
- There is only one major issue that I’d like to raise. The description of results of the crucial in vivo experiment should be expanded with data on the presence of T cells specific to epitopes included into the vaccine. Otherwise, it is hard to justify that the tumor growth inhibition is caused by the antigen-specific immune response and not the non-specific activation of dendritic cells. Alternatively, the DEC-205 mAb should be used as a control.
Response #6: In response to a similar question raised reviewer 2 on the melanoma-antigen specificity of the induced T cell response, we included data in the revised manuscript on the expansion of melanoma antigen-specific T cells in the periphery of immunized tumor-bearing mice (see new Supplementary Figure S5). The gp100- or trp2-specific CD8+T cells were detected in the blood of mice that received the DC-based vaccines containing one or both melanoma epitopes. In addition, DEC-OVA-immunized mice showed no significant increase of gp100- or Trp2-specific CD8+ T cells, indicating that the expansion of T cells in the periphery was dependent upon the DC-based melanoma vaccines. The description of the new data can be found in the results part (lines 361–377) and the discussion (lines 460-470) of the revised manuscript.
Although these data suggest that the DC-based vaccines expand antigen-specific T cells in the periphery of the mice, initiating an anti-tumor immune response, we cannot exclude the presence of other antigen-specific T cells in the tumor microenvironment that could respond to inflammatory stimuli and contribute to the anti-tumor immune responses observed, potentially by epitope spreading. Our analysis of effector and central memory CD4+ T cells in the tumor microenvironment seems to support this idea.
- The potential of therapeutic application of this approach can be limited due to the challenges of manufacturing mAb-like molecules fused with epitope-containing peptide chains. Therefore, data on expression level and protein purity would be useful.
Response #7: For the purpose of this study, we used a transient transfection system to produce the DC-vaccine. The produced DC-vaccines were purified via FPLC using protein G columns and the protein levels were determined by BCA, as described in the methods section. We have included additional figures for the reviewer (Supplementary Figure 1A and Supplementary Figure 3A), which show the analysis of the DC-vaccines by instant blue staining on SDS-Page gel. The staining shows only the 50 kDa and 25 kDa bands corresponding to the heavy and light chain of the antibody, respectively. Additional bands are missing, which confirms that the DC-vaccines were isolated with high purity.
- The use of such vaccines will be limited by patients with particular MHC allele. It’d be interesting to know authors’ ideas on how to deal with this limitation.
Response #8: The reviewer raises an interesting point, which is now added in one sentence to the discussion (lines 472-475). Any strategy to generate off-the-shelf DC-vaccines should take into account not just the expression pattern of antigens by the tumor tissue, but also the MHC alleles of the patients before selecting antigens. Due to the highly polymorphic state of the human HLA locus, thousands of allelic variants have been identified for each locus [1]. However, recent advances in the field have identified criteria that seem to be important for the selection of epitopes for specific HLA alleles [2]. Interestingly, the same study identified similarities between different HLA-I alleles in their antigenic epitope preference. In our study, we used 9-aa peptide sequences and used the flanking sites of the OVA257-264 peptide, which generate a precise proteolytic cleavage. However, the epitope length for HLA-I is not restricted to 9 aa but rather varies from 8-12 aa. The use of a longer aa sequence in a DC-vaccine, or alternative flanking sites, could provide alternative cleavage sites that could be presented on a wider variety of HLA-I alleles. Additionally, and despite the highly polymorphic state of HLA-I alleles, there seem to be predominant alleles that cover the majority of the population [3]. Therefore, different epitopes of the same antigens could be used in DC-vaccines specific for different geographical areas, based on the HLA-I distribution in these areas. Finally, the inclusion of additional MHC-I-restricted epitopes as well as MHC-II-restricted epitopes could not only further boost anti-tumor responses but also compensate for reduced antigen presentation efficiency in fractions of patients that might bare rare HLA-I alleles.
References
- Pishesha, N.; Harmand, T.J.; Ploegh, H.L. A guide to antigen processing and presentation. Nat Rev Immunol 2022, 22, 751-764, doi:10.1038/s41577-022-00707-2.
- Karnaukhov, V.; Paes, W.; Woodhouse, I.B.; Partridge, T.; Nicastri, A.; Brackenridge, S.; Shcherbinin, D.; Chudakov, D.M.; Zvyagin, I.V.; Ternette, N.; et al. HLA variants have different preferences to present proteins with specific molecular functions which are complemented in frequent haplotypes. Front Immunol 2022, 13, 1067463, doi:10.3389/fimmu.2022.1067463.
- Sarkizova, S.; Klaeger, S.; Le, P.M.; Li, L.W.; Oliveira, G.; Keshishian, H.; Hartigan, C.R.; Zhang, W.; Braun, D.A.; Ligon, K.L.; et al. A large peptidome dataset improves HLA class I epitope prediction across most of the human population. Nat Biotechnol 2020, 38, 199-209, doi:10.1038/s41587-019-0322-9.
Round 2
Reviewer 1 Report
Comments and Suggestions for Authors
This reviewer acknowledges that appropriate improvements have been made to the work and that the authors have addressed all issues raised
Reviewer 2 Report
Comments and Suggestions for Authors
Authors' responses have addressed all my concerns. The manuscript can be published in its present form.